# Cell Sheets Restore Secretory Function in Wounded Mouse Submandibular Glands

**DOI:** 10.3390/cells9122645

**Published:** 2020-12-09

**Authors:** Harim T. dos Santos, Kyungsook Kim, Teruo Okano, Jean M. Camden, Gary A. Weisman, Olga J. Baker, Kihoon Nam

**Affiliations:** 1Christopher S. Bond Life Sciences Center, University of Missouri, Columbia, MO 65211, USA; harimtavaresdossantos@health.missouri.edu (H.T.d.S.); CamdenJ@missouri.edu (J.M.C.); WeismanG@missouri.edu (G.A.W.); bakero@health.missouri.edu (O.J.B.); 2Department of Otolaryngology-Head and Neck Surgery, School of Medicine, University of Missouri, Columbia, MO 65212, USA; 3Cell Sheet Tissue Engineering Center (CSTEC), Department of Pharmaceutics and Pharmaceutical Chemistry, University of Utah, Salt Lake City, UT 84112, USA; kyungsook.kim@utah.edu (K.K.); teruo.okano@utah.edu (T.O.); 4Institute of Advanced Biomedical Engineering and Science, Tokyo Women’s Medical University, Tokyo 162-8666, Japan; 5Department of Biochemistry, University of Missouri, Columbia, MO 65211, USA

**Keywords:** cell sheet, extracellular matrix, hyposalivation, regeneration, wound healing

## Abstract

Thermoresponsive cell culture plates release cells as confluent living sheets in response to small changes in temperature, with recovered cell sheets retaining functional extracellular matrix proteins and tight junctions, both of which indicate formation of intact and functional tissue. Our recent studies demonstrated that cell sheets are highly effective in promoting mouse submandibular gland (SMG) cell differentiation and recovering tissue integrity. However, these studies were performed only at early time points and extension of the observation period is needed to investigate duration of the cell sheets. Thus, the goal of this study was to demonstrate that treatment of wounded mouse SMG with cell sheets is capable of increasing salivary epithelial integrity over extended time periods. The results indicate that cell sheets promote tissue organization as early as eight days after transplantation and that these effects endure through Day 20. Furthermore, cell sheet transplantation in wounded SMG induces a significant time-dependent enhancement of cell polarization, differentiation and ion transporter expression. Finally, this treatment restored saliva quantity to pre-wounding levels at both eight and twenty days post-surgery and significantly improved saliva quality at twenty days post-surgery. These data indicate that cell sheets engineered with thermoresponsive cell culture plates are useful for salivary gland regeneration and provide evidence for the long-term stability of cell sheets, thereby offering a potential new therapeutic strategy for treating hyposalivation.

## 1. Introduction

Hyposalivation is the reduction of salivary flow contributing to oral microbial infections that impairs activities of daily living (e.g., speaking, chewing and swallowing) [1]. Hyposalivation is not only associated with radiation therapy, which is administered every year to approximately 65,000 head and neck cancer patients in the United States [2,3], but is also associated with the autoimmune disease Sjögren’s syndrome as well as side effects to certain medications [4]. Current treatments for hyposalivation are limited to the muscarinic receptor agonists pilocarpine and cevimeline that induce saliva secretion from residual acinar cells [4] and the use of saliva substitutes [5]. However, these therapies do not address the underlying cause of hyposalivation, i.e., the destruction of salivary gland acini, and provide only temporary relief [6]. Moreover, pharmacological stimulation of salivary glands may result in systemic adverse effects that limit use in some patients [7]. In light of the above, several alternative treatments for hyposalivation, such as surgical transplantation of stem cells, use of bioengineered organ-germs, growth factor delivery and scaffold implantation, are being developed to improve these patients’ quality of life [8,9,10]. Regarding stem cells, recent studies have shown that mesenchymal-like stem cells from human SMG were able to repair radiation-damaged rat salivary glands [11]. However, stem cell-mediated treatment is limited by cell survival and the relatively short effectiveness of growth factor secreted by stem cells [12,13]. As for bioengineered organ-germ transplantation, previous studies demonstrated that mouse embryonic salivary cells (i.e., those derived from submandibular, sublingual and parotid glands) grown in an organ culture can be transplanted in vivo [9]. However, the modest size of such bioengineered glands and the brief survival period for animal subjects following transplantation significantly decrease the clinical potential of this approach for human application. In terms of growth factors, several studies demonstrated that they maximize ex vivo tissue expansion by regulating salivary gland cell proliferation, regeneration and differentiation [14,15]. However, additional challenges arise from utilizing exogenous growth factors as therapeutics, including low recombinant expression yield, difficulty of purification, high production costs, lack of appropriate delivery methods and unwanted side effects (e.g., tumorigenesis) [16,17,18]. Regarding the use of scaffolds, they allow cell attachment, growth, migration and differentiation [19,20,21,22,23,24,25], although scaffolds need to be optimized for long-term stability and regeneration capacity. Together, these studies indicate an ongoing need for novel technologies that promote salivary gland regeneration and permanent relief from hyposalivation with minimal or no secondary effects. To that end, we introduced the use of cell sheet technology, which consists of plating primary cells on a culture dish covalently covered with a temperature-responsive polymer that enables cell monolayers to completely detach from the dish surface by decreasing the temperature (i.e., from 37 to 25 °C) without enzymatic treatment [26,27]. Because cell sheets are able to maintain extracellular matrix (ECM) and cell–cell junctional proteins, they rapidly attach to the target organs without needing sutures or biomaterials [28,29]. Cell sheets have been successfully used in various animal models and human clinical trials to regenerate multiple tissues, including cartilage, lung, heart, liver, endometrium, intestine, cornea and periodontium [30,31,32,33,34,35,36,37,38,39,40,41,42,43,44,45,46,47,48]. Our recent studies demonstrated that cell sheets promote significant SMG cell differentiation and recovery of tissue integrity in short durational observations [49]. Given that the loss of regenerative ability over time is a long-standing problem in tissue bioengineering, establishing the efficacy of cell sheets in salivary gland regeneration requires demonstration that the cell sheets survive and are functional for a clinically-relevant duration, as addressed in this study.

## 2. Materials and Methods

### 2.1. Materials

TO-PRO-3 iodide, rabbit anti-ZO-1, Alexa Fluor 488 conjugated anti-rabbit IgG secondary antibody and Alexa Fluor 568 conjugated anti-mouse IgG secondary antibody were purchased from Invitrogen (Carlsbad, CA, USA). Rabbit anti-aquaporin 5 (AQP5), mouse anti-cytokeratin 7 (K7) and rabbit anti-TMEM16A were purchased from Abcam (Cambridge, MA, USA). Mouse Na^+^/K^+^-ATPase was purchased from Santa Cruz Biotechnology (Dallas, TX, USA). Mouse anti-E-cadherin was purchased from BD Biosciences (San Jose, CA, USA). Tumor dissociation kit was purchased from Miltenyi Biotec Inc. (Auburn, CA, USA). DMEM/F12, fetal bovine serum (FBS), phosphate buffered saline (PBS), UpCell™ temperature-responsive dishes and strainers (100, 70 and 40 µm) were purchased from Thermo Fisher Scientific (Waltham, MA, USA). Retinoic acid, triiodothyronine, epidermal growth factor (EGF), goat serum, insulin-transferrin-sodium selenite supplement, glutamine, hydrocortisone, pilocarpine, isoproterenol and Tween 20 were purchased from MilliporeSigma (Burlington, MA, USA). Mini-PROTEAN TGX precast electrophoresis gels and tris-glycine SDS running buffer were purchased from Bio-Rad (Hercules, CA, USA). Triton X-100, paraformaldehyde, sodium citrate, xylene, glacial acetic acid, methanol and ethanol were purchased from Fisher Scientific (Waltham, MA, USA). Ketamine and xylazine were purchased from VetOne (Boise, ID, USA). Coomassie Brilliant Blue R-250 was purchased from Genlantis (San Diego, CA, USA). Female 6-week-old C57BL/6J mice (weighing ~15–20 g) were purchased from the Jackson Laboratory (Bar Harbor, ME, USA).

### 2.2. Animals

Our previous studies using SMG cell sheets were performed in female mice. To use a homogeneous experimental model, female 6-week-old C57BL/6J mice were used in this study. Mice were housed in a room with a controlled environment (12-h day/night cycles) and provided with a standard pellet diet and water. Mice were randomized and divided into six groups. Animal studies were performed in compliance with the ARRIVE guidelines and the University of Utah Institutional Animal Care and Use Committee (IACUC; protocol number: 18-03003).

### 2.3. Cell Sheet Preparation

A double layer cell sheet was prepared using a previously described method [49]. Briefly, female 6-week-old C57BL/6J mice were euthanized using CO_2_ followed by abdominal exsanguination. Then, SMG were dissociated using a GentleMACS with a tumor dissociation enzyme mixture. Subsequently, SMG cells were centrifuged at 150× *g* for 5 min at 4 °C and the enzyme mixture was removed. Cells were then re-suspended in DMEM/F12 medium containing 2.5% (*v*/*v*) FBS, 2 nM triiodothyronine, 0.1 μM retinoic acid, 0.4 μg/mL hydrocortisone, 80 ng/mL EGF, 5 ng/mL sodium selenite, 5 mM glutamine, 5 μg/mL insulin and 5 μg/mL transferrin. The cell suspension obtained was then passed through 100, 70 and 40 µm strainers and 1.0 × 10^6^ cells were seeded on FBS-coated 35 mm UpCell™ temperature-responsive dishes, as shown in Figure 1. After eight days of incubation, the first cell sheet was detached from the dish surface by reducing the temperature below 25 °C. After removing culture medium from the dish, a wet transfer membrane was placed over the first cell sheet layer. After 30 min of incubation at room temperature, the cell sheet attached membrane was transferred to a new FBS coated culture dish and incubated at 37 °C for 30 min. Then, the wet membrane was gently removed from the first cell sheet attached culture dish. After making the second cell sheet attached wet transfer membrane in the same way, culture medium was removed from the first cell sheet attached culture dish. Then, the first cell sheet layer on the culture dish (apical side) was covered with the second cell sheet attached wet transfer membrane (basolateral) and incubated at 37 °C for 30 min. Finally, the wet transfer membrane was gently removed from the double layer cell sheet attached dish, which was cultured for an additional day at 37 °C.

### 2.4. Wounded SMG Mouse Model

The wounded SMG mouse model was created following a method reported previously [49,50,51,52]. Briefly, female 6-week-old C57BL/6J mice were anesthetized with 3% (*v*/*v*) isoflurane with an oxygen flow rate of 2.0 L/min. Then, SMG were exposed and surgical wounds created using a 3 mm diameter biopsy punch in both SMGs. Subsequently, wounded SMGs were treated with a double layer cell sheet (experimental group). Mouse SMGs untreated with cell sheets (injured controls) and unwounded SMGs (sham controls) were used as controls.

### 2.5. Histological Studies

After either eight or twenty days, SMGs were dissected and fixed in 4% (***v***/***v***) paraformaldehyde for one day. Then, tissue samples were dehydrated in 70% (***v***/***v***) ethanol, embedded in paraffin wax and cut into 3 μm sections. Tissue sections were deparaffinized with xylene and rehydrated with serial ethanol solutions (twice each with 100%, 95% 80%, 70% and 50%) and distilled water. Next, hematoxylin and eosin stains were performed and specimens were examined using a Leica DMI6000B microscope (Leica Microsystems, Wetzlar, Germany).

### 2.6. Confocal Analysis

The rehydrated specimens were incubated in sodium citrate buffer (10 mM sodium citrate, 0.05% (*v*/*v*) Tween 20, pH 6.0) at 95 °C for 30 min, washed with distilled water and permeabilized with 0.1% (*v*/*v*) Triton X-100 in PBS at room temperature for 45 min. Then, specimens were blocked in 5% (*v*/*v*) goat serum in PBS for 1 h at room temperature and incubated at 4 °C with the following primary antibodies in 5% goat serum overnight: rabbit anti-ZO-1, mouse anti-E-cadherin, rabbit anti-AQP5, mouse anti-K7, rabbit anti-TMEM16A or mouse anti-Na^+^/K^+^-ATPase antibodies. Then, sections were incubated for 1 h with anti-rabbit Alexa Fluor 488 or anti-mouse Alexa Fluor 568 secondary antibodies in 5% goat serum at room temperature. Subsequently, specimens were stained with TO-PRO-3 iodide at room temperature for 15 min at 1:1000 dilutions. Finally, specimens were analyzed using a Zeiss LSM 700 confocal microscope (Carl Zeiss, Oberkochen, Germany), ImageJ and GraphPad Prism 6.

The level of fluorescence intensities was calculated using the following equation:(1)$$\mathrm{Fluorescence}\;\mathrm{positive}\;\mathrm{pixels}\;\left(\%\right)=\;\frac{\mathrm{Number}\;\mathrm{of}\;\mathrm{fluorescence}\;\mathrm{positive}\;\mathrm{pixels}}{\mathrm{Total}\;\mathrm{number}\;\mathrm{of}\;\mathrm{pixels}\;\mathrm{in}\;\mathrm{the}\;\mathrm{image}}\times 100$$

### 2.7. Saliva Flow Rate

Mice were anesthetized with ketamine (100 mg/kg) and xylazine (5 mg/kg) and injected intraperitoneally with pilocarpine (25 mg/kg) and isoproterenol (0.5 mg/kg). Then, stimulated whole saliva was collected using a micropipette for 5 min and saliva flow rates expressed as microliters per minute per gram body weight. Finally, statistical significance was assessed by one-way ANOVA and Dunnett’s post-hoc tests for multiple comparisons to the injured control group.

### 2.8. Saliva Protein Composition

Ten micrograms of saliva samples were denatured at 95 °C for 5 min in a sample loading buffer. Then, ten microliters of denatured sample were loaded onto a 4–15% Mini-PROTEAN TGX precast electrophoresis gel and subjected to electrophoresis in Tris-glycine SDS running buffer at 120 V for 60 min. The electrophoresis gel was fixed with 50% (*v*/*v*) methanol, 40% water and 10% (*v*/*v*) glacial acetic acid for 2 h. Then, the gel was stained with 0.25% (*w*/*v*) Coomassie Brilliant Blue R-250 in 50% (*v*/*v*) methanol and 10% (*v*/*v*) glacial acetic acid for 1 h and destained for 6 h in 20% (*v*/*v*) methanol and 10% (*v*/*v*) acetic acid. Protein images of gels were captured using a Chemi-Doc imaging system (Bio-Rad). Finally, signal intensities of total proteins were analyzed using ImageJ and GraphPad Prism 6.

## 3. Results

### 3.1. Treatment with Cell Sheets Promotes Tissue Organization

Our previous study showed that cell sheets promote wound healing at eight days, as indicated by organized formation of glandular tissue with minimal fibrosis in a wounded SMG mouse model [49]. However, it is unclear whether these effects are maintained over time. For this reason, we investigated whether treatment of wounded mouse SMG with cell sheets promotes wound healing twenty days post-surgery using hematoxylin-eosin staining. As shown in Figure 2a, injured SMG, at eight days post-surgery, show minimal regeneration, Figure 2(a1), loose connective tissue throughout the wound area, Figure 2(a2), and the presence of adipocytes and congested blood vessels at the center of the wound, Figure 2(a3,4), which taken together indicate scar tissue formation, Figure 2(a1–4). As shown in Figure 2b, injured SMG, at twenty days post-surgery, show areas filled with blood vessels, Figure 2(b1–3), adipocytes, Figure 2(b2,3), and vacuolated mononuclear cells, Figure 2(b3,4, green arrows), all of which are consistent with poor wound healing. In contrast, SMG surgical wounds treated with cell sheets show the presence of organized blood vessels throughout the wounded area at eight days post-surgery, Figure 2(c1), and the formation of acinar- and ductal-like structures (Figure 2(c2,3), red arrows) with minimal mononuclear cell infiltrates, Figure 2(c4), early signs of wound healing. Moreover, SMG surgical wounds treated with cell sheets show organized parenchyma (e.g., acinar and ductal structures; Figure 2(d1,2), yellow arrows) at twenty days post-surgery and stroma (e.g., connective tissue and blood vessels; Figure 2(d3,4), blue arrows), comparable to sham controls (Figure 2e,f). These results indicate that cell sheets promote tissue organization as early as eight days after transplantation and that these effects endure through Day 20.

### 3.2. Treatment with Cell Sheets Promotes Epithelial Polarization

Our previous study showed that cell sheets polarize early after transplantation in SMG (i.e., eight days post-surgery) [49]. Because it remains unclear whether the tissue regenerative effects of cell sheets are retained or enhanced over time, we further investigated whether treatment with cell sheets increases the expression of the apical tight junction-associated protein zonula occludens-1 (ZO-1) and the basolateral adhesion protein E-cadherin, consistent with late-stage polarization. As shown in Figure 3a, injured SMG at eight days post-surgery show an absence of polarity markers, while injured SMG at twenty days post-surgery (Figure 3b) show disorganized expression of ZO-1 (white arrows) and absence of E-cadherin, consistent with the lack of epithelial integrity. Conversely, SMG surgical wounds treated with cell sheets (Figure 3c) show expression of apical ZO-1 (green arrows) and basolateral E-cadherin (red arrows) eight days post-surgery with only minimal areas lacking expression of these markers (yellow dotted line), results consistent with early stages of epithelial polarization. Moreover, SMG surgical wounds treated with cell sheets (Figure 3d) show expression of apical ZO-1 (green arrows) and basolateral E-cadherin (red arrows) twenty days post-surgery, comparable to sham controls (Figure 3e,f) and consistent with advanced stages of epithelial polarization. Furthermore, a pixel quantification analysis of the polarity markers from confocal images indicates that cell sheet transplantation in wounded SMG increases ZO-1 and E-cadherin fluorescence intensities at eight and twenty days post-surgery, compared to injured controls (Figure 3g,h), indicating a significant time-dependent enhancement of cell polarization.

To determine whether cell sheet transplantation promotes cell differentiation, we investigated the presence of acinar and ductal markers [53]. As shown in Figure 4a,b, injured SMG at eight and twenty days post-surgery display an absence of both AQP5 (water transporter, an acinar apical marker) and K7 (a ductal apical marker), indicating a lack of cell differentiation. In contrast, SMG surgical wounds treated with cell sheets (Figure 4c) show a scattered expression of AQP5 (green arrows) and K7 (red arrows) at eight days post-surgery, indicating early-stage differentiation. Additionally, SMG surgical wounds treated with cell sheets (Figure 4d) demonstrate strong expression of AQP5 (green arrows) and K7 (red arrows) at twenty days post-surgery, similar to sham controls (Figure 4e,f) and consistent with advanced-stage differentiation. Furthermore, a pixel quantification analysis of the differentiation markers from confocal images indicates that cell sheet transplantation in wounded SMG increases AQP5 and K7 fluorescence intensities at eight to twenty days post-surgery, as compared to injured controls (Figure 4g,h), indicating a significant time-dependent enhancement of cell differentiation.

To determine whether treatment of wounded SMG with cell sheets increases ion transporter expression, we detected the transmembrane member 16A (TMEM16A) and the sodium-potassium pump (Na^+^/K^+^-ATPase) proteins, both of which are essential for fluid secretion [54,55]. As shown in Figure 5a, injured SMG at eight days post-surgery lack both TMEM16A and Na^+^/K^+^-ATPase. Likewise, injured SMG at twenty days post-surgery show an absence of TMEM16A and disorganized expression of Na^+^/K^+^-ATPase (Figure 5b, white arrows). Conversely, SMG surgical wounds treated with cell sheets (Figure 5c) displayed a scattered expression of apical TMEM16A (green arrows) and a strong expression of basolateral Na^+^/K^+^-ATPase (red arrows) at eight days post-surgery. Moreover, SMG surgical wounds treated with cell sheets (Figure 5d) show strong TMEM16A and Na^+^/K^+^-ATPase fluorescence intensity at twenty days post-surgery, comparable to sham controls (Figure 5e,f). Furthermore, a pixel quantification analysis of TMEM16A and Na^+^/K^+^-ATPase from confocal images shows increases in TMEM16A and Na^+^/K^+^-ATPase fluorescence intensities between eight and twenty days post-surgery, as compared to injured controls (Figure 5g,h), indicating time-dependent increases in ion transporter expression in response to cell sheet transplantation in SMG.

### 3.3. Treatment with Cell Sheets Promotes Secretory Function

We measured salivary flow rates at eight and twenty days post-surgery to determine whether the SMG exhibits secretory function. The results indicate that injured SMGs treated with cell sheets show a significant increase in saliva flow rates at eight and twenty days post-surgery compared to injured SMGs (Figure 6a). Next, we determined the protein composition using SDS-PAGE of stimulated saliva at eight and twenty days post-surgery in injured mice treated with cell sheets, indicating higher levels of salivary proteins compared to injured controls (Figure 6b), with levels of 10–30 kDa proteins significantly enhanced (Figure 6c). Together, these results indicate that cell sheet transplantation in wounded mouse SMG enhances saliva quantity and protein composition at both eight and twenty days post-surgery, whereas salivary protein composition at twenty days post-surgery with cell sheets was approaching sham controls (Figure 6c, dashed line).

## 4. Discussion

Our recent studies demonstrated that cell sheets promote tissue healing and secretory function in a wounded mouse SMG at eight days post-surgery [49]. Here, we extended the regeneration studies beyond this time point to determine that the regenerative effects of cell sheets identified herein were further improved at twenty days post-surgery and were comparable to sham controls. In fact, the effects of cell sheets observed at twenty days post-surgery include the following: (a) enhanced attachment to the transplantation site and wound healing (Figure 2); (b) increased epithelial integrity (Figure 3, Figure 4 and Figure 5); and (c) robust secretory function (Figure 6), as compared to injured controls without cell sheets. Regarding attachment, cell sheets directly bind to the wounded gland immediately after transplantation (Appendix A
Appendix A). We believe that this effect is due to the presence of ECM proteins on the cell surface [56,57], consistent with previous studies showing that expression of fibronectin and collagen in mesenchymal and dental pulp stem cell sheets enhanced attachment and promoted dental pulp formation, both in vitro [58] and in vivo [59,60]. Moreover, these ECM proteins were upregulated in mesenchymal stem cell sheets with enhanced cell attachment, thereby promoting spinal cord injury repair in a rat model [61]. Other ECM proteins, such as proteoglycans and type I and type III collagens, are expressed in mesenchymal stem cell sheets and enhance bone-tendon healing in a rabbit model [62]. Such studies suggest that cell sheets promote expression of ECM proteins, thereby leading to increased cell attachment to an injured site, which in turn serves to facilitate regenerative responses through this anchoring effect.

Regarding wound healing, we observed that treatment with SMG cell sheets enabled complete engraftment to the injured site, formation of well-organized blood vessels, minimal inflammation and low fibrosis, Figure 2(d3,d4). These results indicate that cell sheets promote rapid wound healing, consistent with previous findings with other organs. Specifically, a wounded skin mouse model treated with mesenchymal stem cell sheets formed organized blood vessels and exhibited increased expression of angiogenic factors, thus leading to tissue healing at twenty-one days post-surgery [63]. Likewise, mesenchymal stem cell sheets were shown to improve kidney function by reducing fibrosis and microvasculature damage in a renal ischemia-reperfusion injury rat model fourteen days post-surgery [64]. Furthermore, mesenchymal stem cell sheets have been shown to restore function, while decreasing fibrosis in a chronic liver injury mouse model nine days after surgery [65]. These studies demonstrate the utility of cell sheets for enhancing tissue repair, while at the same time limiting fibrotic growth, which has consistently proven to be a major challenge for tissue engineering targeting long-term growth for clinical applications.

Regarding epithelial integrity, cell sheets have been shown to enhance the expression of polarity and differentiation markers [66,67,68], both of which are indicative of intact tissue. Specifically, our previous studies demonstrated that salivary gland cell sheets exhibit enhanced apical expression of the tight junction-associated protein ZO-1, both in vitro and in vivo, thereby leading to increases in the intracellular calcium concentration and saliva secretion, respectively [49]. In the current study, both polarization and secretory function were enhanced at twenty days after cell sheet transplantation in injured SMG, compared to the injured control groups (Figure 3, Figure 4, Figure 5 and Figure 6). These findings are consistent with previous studies using retinal pigment epithelial cell sheets, which improved both epithelial polarization and barrier function in a rat model of inherited retinal degeneration [66]. Furthermore, transplantation of retinal epithelial cell sheets has been noted to increase tight junction formation in damaged retina, whereas a simple cell suspension treatment neither formed tight junctions nor promoted tissue repair [67,69]. Likewise, results of the current study showed that SMG cell sheets exhibit enhanced expression of the ion transporter TMEM16A (Figure 5), which is an early contributor to agonist-induced fluid secretion [70]. Similarly, Na^+^/K^+^-ATPase maintains the osmotic balance of cells by exporting Na^+^ and importing K^+^ at the expense of adenosine 5′-triphosphate (ATP) hydrolysis, which is critical for the formation of ion gradients that regulate saliva secretion [71]. We postulate that the enhanced expression of these ion transporters (Figure 5) explains the improved saliva secretion caused by cell sheet transplantation in injured mouse SMG at twenty days post-surgery (Figure 6a), which is consistent with earlier studies using human corneal endothelial cell sheets in which enhanced Na^+^/K^+^-ATPase expression was linked to improved secretory function [72]. To confirm that the regenerated area within the cell sheet-treated injured SMG at twenty days post-surgery was indeed composed of salivary gland tissue, we observed increased expression of the acinar differentiation marker AQP5 and the ductal differentiation marker K7 (Figure 4). These results indicate that the repaired tissue was comprised of acinar and ductal cells, consistent with other cell sheet studies where cell differentiation was observed [73,74,75]. Of note is the significant enhancement observed in saliva secretion at both eight and twenty days post-surgery, as compared to injured SMG, with saliva protein composition at twenty days post-surgery approaching sham controls (Figure 6). Specifically, we observed that cell sheet-treated mouse SMG at twenty days post-surgery exhibited enhanced expression of small molecular weight salivary proteins, which are abundant in healthy saliva, promote enamel re-mineralization and provide protection against periodontal disease through protease inhibition in oral bacteria [76,77].

The above studies indicate that cell sheets promote epithelial integrity by increasing tight junction-associated protein expression, which in turn leads to enhanced saliva production [78,79]. Beyond fluid secretion, epithelial integrity is associated with organized secretion of paracrine factors due to cell polarization. For instance, studies using permeable supports in vitro have shown that polarized cell sheets secrete growth factors bi-directionally (i.e., apical secretion of pigment epithelium-derived factor (PEDF) and basolateral secretion of VEGF [66]), similar to secretion patterns observed in healthy tissues in vivo and contrasting with the disorganized expression of these factors observed with cell suspensions [80]. Moreover, enhanced expression of stromal cell-derived factor 1 (SDF-1), hepatocyte growth factor (HGF) and VEGF were observed following a myoblast cell sheet implantation in a rat coronary artery ligation model, compared to cell suspension only [81]. Compared to cell suspensions, mesenchymal stem cell sheets exhibited upregulated expression of interleukin-10 (IL-10), vascular cell adhesion molecule-1 (VCAM-1), metallopeptidase inhibitor-1 (TIMP-1), insulin-like growth factor-1 (IGF-1), matrix metalloproteinase-2 (MMP-2), hypoxia-inducible factor 1-α (HIF1-α), SDF-1, matrix metallopeptidase-9 (MMP-9) and basic fibroblast growth factor (bFGF) and enhanced wound healing in a rat coronary artery ligation model [82]. Collectively, these results suggest that cell sheet polarization leads to organized paracrine factor secretion, thereby enabling formation of mature blood vessels and immune cell infiltration that stimulates clearance of debris and promotes wound healing, which demonstrates that cell sheets represent a significant improvement on the current use of cell suspensions to repair damaged tissues.

We demonstrated that small molecular weight protein levels were similar to sham controls at twenty days post-transplantation of salivary cell sheets in injured SMG, thereby extending our previous findings derived at eight days and showing that not only saliva quality but also saliva composition and wound healing are enhanced over time. These results were obtained without the need for external biomaterials and/or stem cells, both of which may cause undesirable secondary effects. The current study adds to a growing body of literature demonstrating the impressive wound healing qualities of cell sheets in the repair of salivary glands and other tissues. Nonetheless, these findings should be extended to tissue regeneration in humans, which has not been sufficiently investigated, and a better understanding of the mechanisms underlying cell sheet-mediated tissue repair should help to optimize this approach for repair of salivary glands in patients undergoing radiation therapy for head and neck cancer.

This study focused on responses in female mice rather than using both sexes. It is well-established that sexual dimorphism occurs in mouse SMG (e.g., the rate of glandular development [83] and cell types [84]). Moreover, previous studies have shown that male mice display higher expression levels of growth factors than their female counterparts [85]. A recent study in our lab demonstrated that SMG wound healing occurs at different rates in male and female C57BL/6J mice [86]. There are also major structural and physiological differences between male and female SMGs that are likely relevant to the repair of irradiated [87] or damaged [52] salivary glands. Therefore, it will be important to identify the mechanisms underlying sex-dependent differences in salivary gland regeneration patterns in future studies to define the effects of cell sheets on SMG wound healing in male and female mice and humans.

## Figures and Tables

**Figure 1 cells-09-02645-f001:**
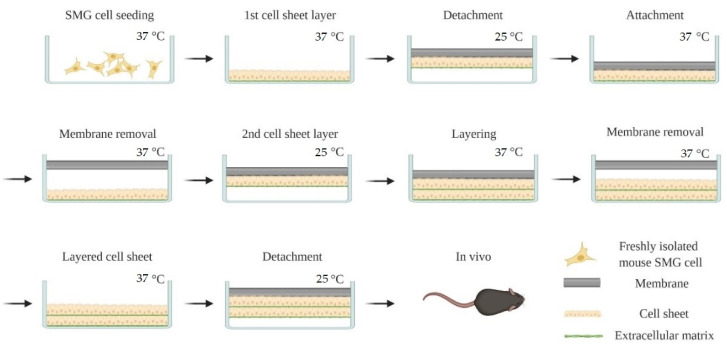
Layered cell sheet preparation process. Created with BioRender.com.

**Figure 2 cells-09-02645-f002:**
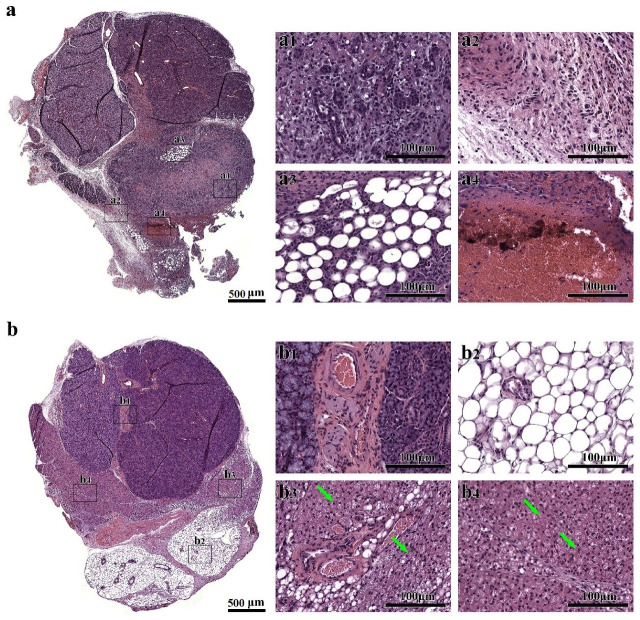
Cell sheets promote wound healing through twenty days post-transplantation. Shown are post-injured SMG ((**a**) Day 8 or (**b**) Day 20), wounded SMG treated with cell sheets ((**c**) Day 8 or (**d**) Day 20) and sham controls ((**e**) Day 8 or (**f**) Day 20). Tissue sections were stained with hematoxylin–eosin and analyzed using a Leica DMI6000B microscope. Green arrows indicate vacuolated mononuclear cells, red arrows indicate acinar- and ductal-like structures, yellow arrows indicate parenchyma and blue arrows indicate stroma.

**Figure 3 cells-09-02645-f003:**
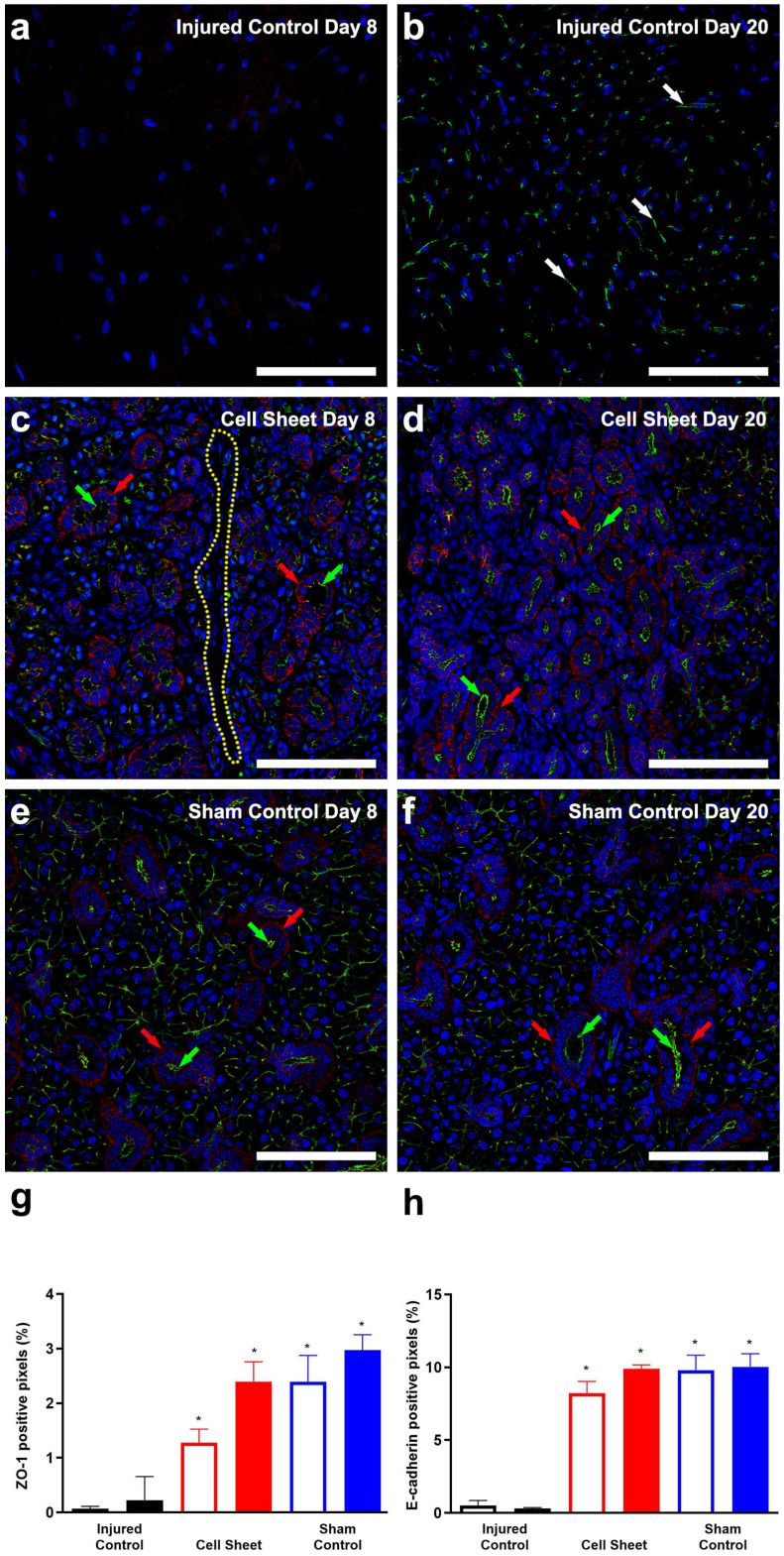
Cell sheets promote epithelial polarization: injured SMG (**a**,**b**); injured SMG treated with cell sheets (**c**,**d**); and sham controls (**e**,**f**). Tissue samples were incubated with rabbit anti-ZO-1 (**a**–**f**) (green) and mouse anti-E-cadherin (**a**–**f**) (red) antibodies and analyzed using a Zeiss LSM 700 confocal microscope, with data representative of results from five experiments and white bars indicating 100 µm. Positive areas of ZO-1 (**g**) and E-cadherin (**h**) fluorescence were captured using ImageJ software. Data are expressed as means ± SD of results from five independent experiments, where statistical significance was assessed by one-way ANOVA (*p* < 0.05) and Dunnett’s post-hoc test for multiple comparisons to injured controls (open bars: Day 8; closed bars: Day 20); * significant difference from injured controls. White/green and red arrows indicate ZO-1 and E-cadherin expression, respectively. Finally, a yellow dotted line indicates a lack of ZO-1 and E-cadherin expression.

**Figure 4 cells-09-02645-f004:**
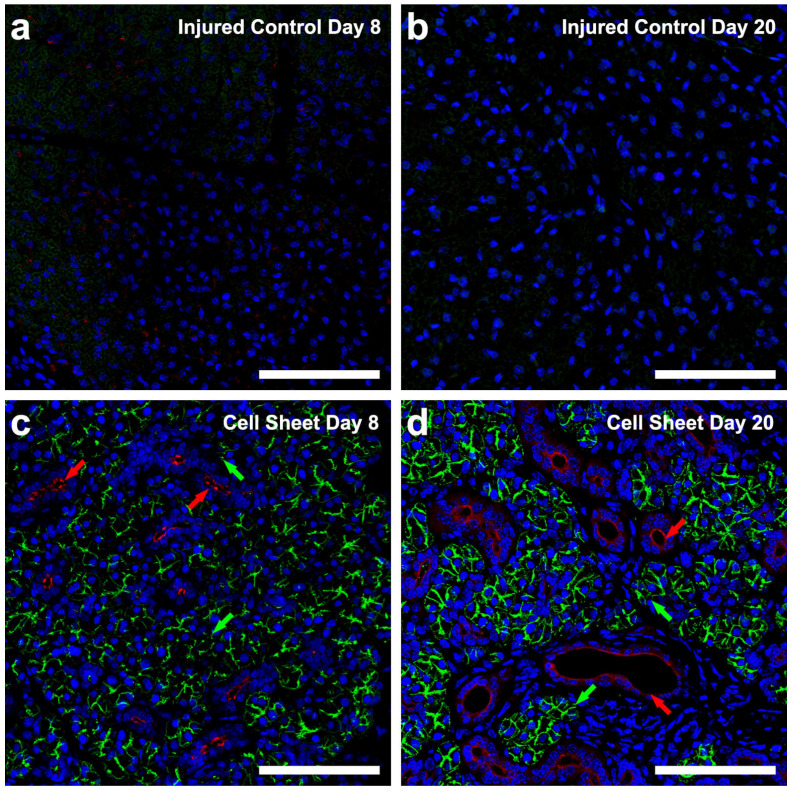
Cell sheets enhance expression of differentiation markers: injured SMG (**a**,**b**); injured SMG treated with cell sheets (**c**,**d**); and sham controls (**e**,**f**). Tissue samples were incubated with rabbit anti-AQP5 (**a**–**f**) (green) and mouse anti-K7 (**a**–**f**) (red) antibodies and analyzed using a Zeiss LSM 700 confocal microscope, with data representing results from five experiments and white bars indicating 100 µm. Positive areas of AQP5 (**g**) and K7 (**h**) fluorescence were captured using ImageJ software. Data are expressed as means ± SD of results from five independent experiments, where statistical significance was assessed by one-way ANOVA (*p* < 0.05) and Dunnett’s post-hoc test for multiple comparisons to injured controls (open bars: Day 8; closed bars: Day 20); * significant difference from injured controls. Green and red arrows indicate AQP5 and K7 expression, respectively.

**Figure 5 cells-09-02645-f005:**
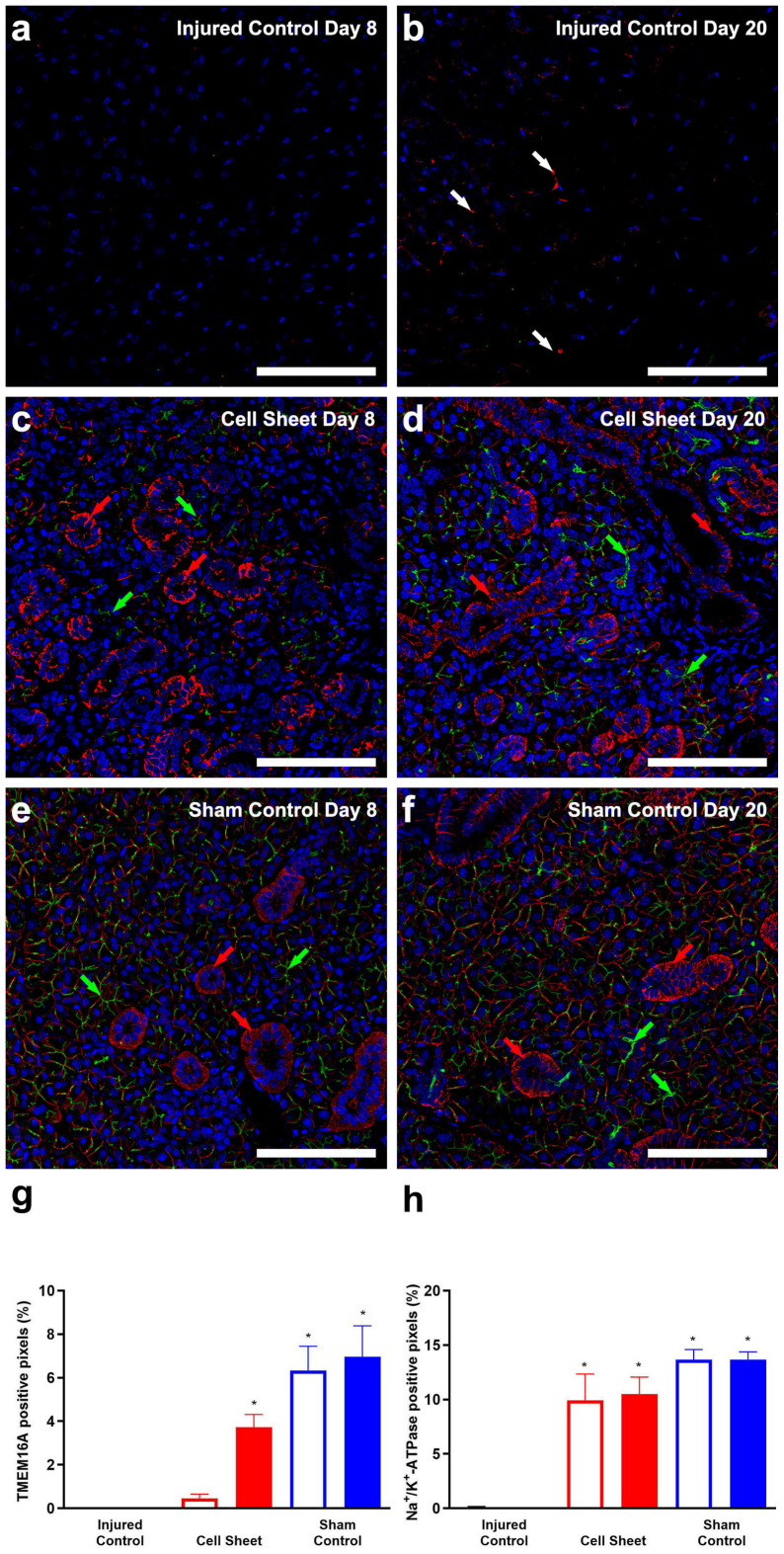
Cell sheets promote expression of ion transporters: injured SMG (**a**,**b**); injured SMG treated with cell sheets (**c**,**d**); and sham controls (**e**,**f**). Tissue samples were incubated with rabbit anti-TMEM16A (**a**–**f**) (green) and mouse anti-Na^+^/K^+^-ATPase (**a**–**f**) (red) antibodies and analyzed using a Zeiss LSM 700 confocal microscope, with data representing results from five experiments and white bars indicating 100 µm. Positive areas of TMEM16A (**g**) and Na^+^/K^+^-ATPase (**h**) fluorescence were captured using ImageJ software. Data are expressed as means ± SD of results from five independent experiments, where statistical significance was assessed by one-way ANOVA (*p* < 0.05) and Dunnett’s post-hoc test for multiple comparisons to injured controls (open bars: Day 8; closed bars: Day 20); * significant difference from injured controls. Green and white/red arrows indicate TMEM16A and Na^+^/K^+^-ATPase expression, respectively.

**Figure 6 cells-09-02645-f006:**
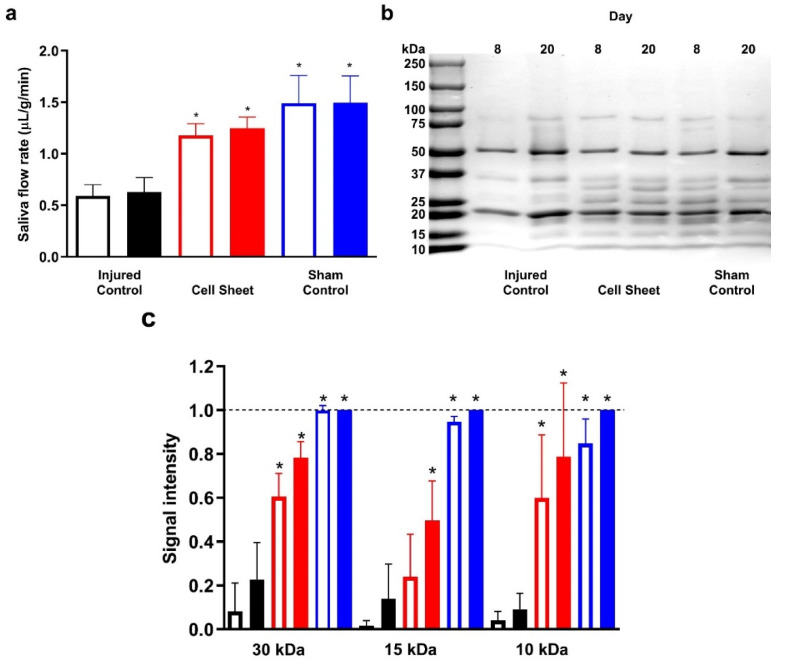
Treatment with cell sheets preserves salivary secretory function. (**a**) As indicated (black bars: injured controls; red bars: cell sheet treatment; blue bars: sham controls) mice were anesthetized and stimulated with pilocarpine and isoproterenol, then saliva was collected for 5 min to determine flow rate. Data are expressed as means ± SD of results from seven independent experiments, where statistical significance was assessed by one-way ANOVA (*p* < 0.01) and Dunnett’s post-hoc test for multiple comparisons to injured controls (open bars: Day 8; closed bars: Day 20). (**b**) Saliva protein was fractionated by SDS-PAGE and stained with 0.25% (*w*/*v*) Coomassie Brilliant Blue R-250 to assess salivary protein patterns in whole saliva and (**c**) signal intensities were compared to injured controls and analyzed using both ImageJ and GraphPad Prism 6. Data are expressed as means ± SD of results from three independent experiments, where statistical significance was assessed by one-way ANOVA (*p* < 0.05) and Dunnett’s post-hoc test for multiple comparisons to injured controls at Day 20; * significant difference from injured controls.

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
