# Peer review of "Cell Sheets Restore Secretory Function in Wounded Mouse Submandibular Glands"

_cells, 2020, doi:10.3390/cells9122645_

Round 1

Reviewer 1 Report

The work of dos Santos et al. is a continuation of a previous work published by the same group where the showed that wounded mouse submandibular glands restore gland function when treated for 8 days with cultured cells isolated from mouse submandibular glands.

In the present study, the authors showed that 20-day treatment regenerate gland tissue and significantly restore gland function. Submandibular gland phenotype is significantly improved at the 20th day compared to that displayed at the 8th day. 

The experiments are well done and the manuscript is well written. I have one major concern:

1) I am wondering why only female mice were used in the study. Did the authors performed experiments with male mice? In the case that no experiments were performed in males, the authors should explain in the manuscript why the study was only conducted with female mice.

Minor concerns:

  1) Figure 1 requires modification. Structures are very difficult to identify in magnifications (for instance Fig 1a1 and Fig 1b3). I recommend to show higher magnifications and increase the size of the magnifications.

2) In the figure legends, it is shown that * denotes statistical significance. However, it is not clear if the significance is respect to sham controls. Please modify figure legends to clarify this issue.

3) In lines 269-270, it is stated that "  Moreover, saliva of mice whose wounded SMG were treated with cell sheets displayed a significantly lower expression of PRPs and cystatins when compared to sham controls at eight days post-surgery (Figure 5c) ". However, * showing statistical significance in Fig 5c is missed. I am wondering if the * was omitted by mistake or there is no significance in the protein levels between the different groups. In case there is no significance, I recommend to remove Figs 5b and 5c.

I recommend the manuscript acceptance after addressing the concerns raised above.

Author Response

We thank the reviewer for providing valuable advice that we have used to improve the manuscript. Following are our specific responses to the reviewer’s concerns. Please see the attachment.

Reviewer 2 Report

The authors have submitted a manuscript entitled " Cell Sheets Restore Secretory Function in Wounded 2 Mouse Submandibular Glands " and described the effects of cell sheets over 20 days for tissue repair of submandibular salivary glands.

The manuscript is well written, clear to follow and results are presented according to the story. Anyhow, there are some concerns:

Introduction:

Line 41: Some reasons for hyposalvation are missing (Sohgren syndrome, polypharmacology,..), please some more.

Materials & Methods:

  • Approval number for animal experiments is missing and a link to the “strict” guidelines, otherwise it is not reproducible under which conditions the animal experiments have been carried out.
  • Please add the companies of the purchased materials in section “2. Materials and Methods”.
  • If possible, simplify the cell sheet preparation in 2.2, e.g. by a graphical demonstration.
  • Line 109: change both glands to both SMG glands.
  • Line 133: in how far does the anesthesia changes the saliva flow rate ?
  • Line 146: How were the proteins PRPs and cystatins really proven. The reviewer knows that it is a common technique to analysis only stained SDS_PAGE gels, but this is no proof o the proteins, western blotting and antibody staining should be done at least once to confirm the identify of the bands.
  • Line 183 and others: please note that ZO-1 is NO tight junction protein, it is a tight junction associated protein since ZO-1 is not interacting with trans claudins to seal the paracellular clefts.

Results:

  • For a better presentation of the results, a larger image for Figure 1x1-4 would be of advantage
  • Regarding the Figure 2-4, the treated mouse SMGs show similar expression levels of the demonstrated proteins as the sham controls (or in Figure 4 even higher expression). Hence it would be of advantage to emphasize the similarity with similar arrows as the treated mouse SMGs samples.
  • Fig3: it seems that there is an overlay of images.
  • For Figure 3-4, please explain why the ANOVA tests were only performed against sham controls on day 20 and not separately for day 8 and 20 as in Figure 5?
  • In general, it would be recommended to exchange “untreated control” by “injuryed control” throughout the total manuscript, then it is immediately clear that the control is an untreated injured control.
  • Fig 5: It seems that several data points of the 8 days data points are very similar to the first paper of the authors about their cell sheet technology (see [49]). Therefore, in case the authors have used the already published data points (and just added one additional data point in order to achieve 7 instead of n=6 in their first publication), then it is highly recommended to state that the majority of day 8 data of figure 5 have already been published in [49].
  • Figure 5b: The gel is artificially brightened, the authors should present the “real” manifold of bands of the gel. Moreover, authors should provide protein concentration data of the applied saliva samples on the gels.  
  • Regarding the results presented in Figure 5, the reviewer does not recognize similar levels of PRP as described in text in section 3.3 for day 20 between sham controls and treated SMG as the quartiles of the boxplots do not overlap (according to the graph the median is around 0.6 ±0.2 while the sham control is set to 1.0.)

Discussion:

  • In line 242, 330 minor formatting mistakes can be found in the text.
  • Line 320: Why have the authors not analysed a “real” tight junction protein, such as a tissue-specific claudin.
  • Line 367: The authors have not shown PRPs nor cystatin, since they have not proven their presence by e.g. antibodies, they just showed some unidentified protein bands (although this Is believed to be state-of-the-art).

Why is the supplementary figure not presented in a sole file ?

In summary, the publication shows that the previously developed cell sheet technology is able to be applied for 20 days instead of previously published eight days, and that this prolonged treatment improves the outcome with regard to the regenerated salivary tissue. Major questions like why they have not tried to translate their technique to human cells are not targeted, but also not discussed in this manuscript.

Author Response

(The authors gave the same response as above.)
